# The Effects of Temperature and Salinity Stressors on the Survival, Condition and Valve Closure of the Manila Clam, *Venerupis philippinarum* in a Holding Facility

Hyeonmi Bae [1,2], Jibin Im [1,2], Soobin Joo [1,2], Boongho Cho [1,2] and Taewon Kim [1,2,*]

[1] Department of Ocean Sciences, Inha University, 100 Inha-ro, Michuhol-gu, Incheon 22212, Korea; qogusala@gmail.com (H.B.); dlawlqls2307@gmail.com (J.I.); soobinism94@gmail.com (S.J.); boonghocho@gmail.com (B.C.)

[2] Program in Biomedical Science and Engineering, Inha University, 100 Inha-ro, Michuhol-gu, Incheon 22212, Korea

* Correspondence: ktwon@inha.ac.kr; Tel.: +82-10-8726-3070

**Abstract:** We investigated the response of the Manila clam *Venerupis philippinarum* to possible temperature and salinity changes in a holding facility. First, clams were exposed to four temperatures for 15 days. Valve closure and survival of clams exposed to seawater at 18 °C were higher than that of those exposed to seawater at 24 °C. Second, clams were exposed to six salinities for 15 days. Survival of clams exposed to two salinity fluctuation conditions (24–30 and 27–24 psu) was lower than that of clams exposed to constant 30 psu conditions. Valve closures of clams exposed to constant low salinity conditions (24 psu) and two salinity fluctuation conditions (24–30 and 27–24 psu) were higher than those exposed to constant 30 psu conditions. Lastly, clams were exposed to two different temperatures and three different salinity conditions for 8 days. Valve closure and survival decreased significantly under the combination of 24 °C and 18 psu. These results suggest that an increase in temperature or a wider range of salinity fluctuations are detrimental to the survival of the Manila clam. The synergistic effect of temperature and salinity stressors may decrease the survival period of clams compared to the effect of a single stressor.

**Keywords:** warming; freshening; salinity fluctuation; multiple stressors; stock holding

## 1. Introduction

Increased anthropogenic greenhouse gas emissions have contributed to climate-related changes, such as global warming and extreme weather events, according to the Intergovernmental Panel on Climate Change report [1]. The global mean surface temperature is predicted to increase by 1.0–3.7 °C by 2100 [2]. In addition, the number of melting glaciers, the frequency and intensity of heavy rainfall events, and the input of freshwater from rivers may increase, due to global warming [3,4]. These events can affect the marine ecosystem, especially through direct effects on marine organisms in the coastal zone [5,6].

Estuaries are among the most ecologically and economically important environments and are sensitive to the effects of climate change [7,8]. Temperature and salinity are important environmental factors affecting the survival of marine benthic organisms, and increasing temperatures and decreasing salinity can act as dominant physical stressors [9]. The reported effects of an increase in temperature include changes in growth [10,11], behavior [12,13], metabolism [14,15], and mortality [12,16]. Salinity effects include abnormal endogenous rhythms [17], changes in behavior [18–20], decreased condition index [21,22], and increased mortality [23]. Short-term temperature and salinity stressors, such as heatwaves and heavy rainfall, can threaten growth, recruitment, and survival at the population level [24,25]. To date, there have been many studies on the single-stressor effects, but studies on the effect of combined stressors are limited. The combination of temperature and salinity effects results in reduced energy [26], and increased mortality [27]. The

combination of multiple stressors may induce both additive, synergistic, or antagonistic responses [16,28,29]. Therefore, it is important to understand the effects of both single and combined stressors to estimate the ecological impact of climate change on marine benthic animals and economic damage in aquaculture [30,31].

Bivalves are valuable biological resources in estuaries and are generally harvested when they grow to market size [9]. Once harvested, fresh bivalves are transported to the local market. However, extreme weather conditions and unforeseen problems may delay the timely transport of bivalves to the market, and bivalves can spend extended time in holding facilities [32,33]. In these situations, bivalves can face fluctuating physical stressors during the holding period [34,35]. Because the water used at the holding facilities is pumped up from the surrounding bay, the bivalves in the facility may be exposed to similar temperature and salinity conditions as those in the field [35]. Several experimental studies have investigated the influence of exposure to fluctuating physical stressors on the survival of bivalves during the holding period, but most of them focused on mussels, which are epifaunal species with a high frequency of extended exposure to the air [32–36]. Conversely, studies on clams, which are infaunal species with a relatively low frequency of exposure to the air, are very rare.

In consideration of this, we investigated the response of the Manila clam, *Venerupis philippinarum*, to possible temperature and salinity changes in a holding facility. *V. philippinarum* is distributed in the northwest Pacific coasts, such as Korea, Japan, China, and the Philippines, and has been introduced to Europe and North America [37,38]. *V. philippinarum* is an infaunal suspension-feeding bivalve with high ecological and commercial importance [37,39,40].

We hypothesized that the impact of climate-related events would negatively affect the survival and behavior of *V. philippinarum* in the holding facility. Three experiments were conducted to test this hypothesis. First, to determine the effect of increasing temperature, the clams were exposed to four treatments of varying temperatures under constant salinity. Second, to determine the effect of decreasing salinity and fluctuations in salinity, clams were exposed to six different salinity treatments under constant temperature. Lastly, to determine the effect of the combined temperature and salinity changes, clams were exposed to two different temperatures and three different salinity conditions.

## 2. Materials and Methods

### 2.1. Collection and Acclimation

For Experiment 1, *V. philippinarum* were collected from Gunpyeong-ri in the Yellow Sea, Korea (37°07′05.1″ N, 126°36′47.5″ E). For Experiment 2 and Experiment 3, *V. philippinarum* that were originally cultured from Muui-Do in the Yellow Sea, Korea (37°23′31.7″ N, 126°24′56.0″ E) were purchased from a local market located in Incheon, Korea (37°27′13.6″ N, 126°36′22.4″ E). The collected clams were transported to the laboratory in an icebox. In the laboratory, the clams were placed in six acclimation tanks (60 L, 38 cm × 55 cm × 30 cm, W × L × H) without added food for 7–8 days to maintain the same gut condition of individuals before the experiment. The water was recirculated using a filter media and a pump. After acclimation, they were individually marked using a paint marker (Kangnam Kpi Co., Seoul, Korea), and their shell lengths were measured with digital calipers (CD-15PSX, Mitutoyo Corp., Kanagawa, Japan) to the nearest 0.01 mm. Water temperature and salinity were measured using a YSI pro2030 (Yellow Spring Instruments Inc., OH, USA) twice every day. The water temperature at each treatment was maintained during the experiment using a water bath and heaters (BS-3000, Dong-woo Electric, Daegu, Korea), while salinity at each treatment was attained by adding artificial sea salt (Red Sea, HO, USA).

### 2.2. Experimental Setup

In each experiment, large experimental chambers (126 L, 75 cm × 80 cm × 20 cm, W × L × H) were used and continuously aerated using an air pump (YP-15A, Youngnam, Busan, Korea). In each chamber, water was pumped through a filter (EHEIM Professionel

4 + 250, EHIM GmbH and Co. KG, Deizisau, Germany) and one cooler (DAEIL DBC-150, Daeil, Busan, Korea) (Figure 1a) and then entered via a hose to each jar containing clams in a submerged net pot (Figure 1b). The water of the jars was overflowed into each chamber and circulated. The photoperiod followed a 12/12-h light/dark cycle.

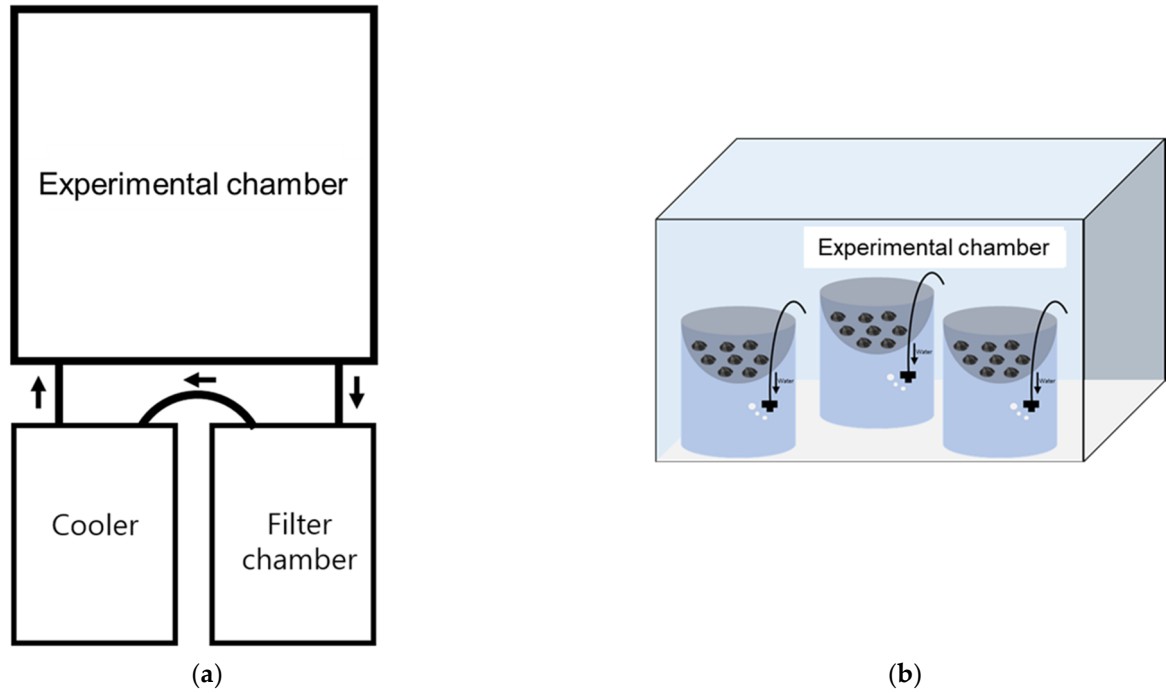

<div align="center">(<b>a</b>)          (<b>b</b>)</div>

**Figure 1.** (**a**) Diagram of the experimental setup and (**b**) diagram of each experimental chamber.

### 2.2.1. Temperature (Experiment 1)

To investigate the effect of temperature on the survival and behavioral response of *V. philippinarum*, the first experiment was conducted from 21 September to 5 October 2018. A total of 360 clams [shell length (SL) $33.51 \pm 0.095$ mm: mean $\pm$ standard error (SE)] were randomly selected from the acclimation tank of each treatment. Acclimation tanks were maintained at 15 °C on the first day and then increased by 1.5 °C per day to the experimental conditions for each treatment for seven days. ($T = 15$ °C, 18 °C, 21 °C, and 24 °C; Sal = 30 psu). Nine clams were allocated to each replicate. Ten replicates per treatment were placed inside each experimental chamber. The clams were maintained under four temperature treatments (15 °C, 18 °C, 21 °C, and 24 °C) for 15 days according to the regular seawater temperatures recorded between May and August at the rearing site (Sea Grant Gyeonggi, Korea). An ambient salinity of 30 psu was maintained throughout all four experimental conditions.

### 2.2.2. Salinity (Experiment 2)

To investigate the effect of different salinity conditions (low salinity and salinity fluctuations) on *V. philippinarum*, the second experiment was conducted from 12 June to 26 June 2020. A total of 384 clams (SL $33.52 \pm 0.091$ mm: mean $\pm$ SE) were randomly selected from the acclimation tank, which maintained steady conditions for seven days. ($T = 21$ °C; Sal = 30 psu). Eight clams were allocated to each replicate. Eight replicates per treatment were placed inside the experimental chamber (Figure 2a). The clams were maintained under six experimental conditions for 15 days. We exposed clams to the following experimental treatments at a constant temperatures (21 °C): (1) Constant salinity (24,27, and 30 psu) and (2) fluctuating salinity (24–30, 27–24, and 30–27 psu). These salinity ranges correspond to those experienced by Manila clams in the holding tank during the summer season (Sea Grant Gyeonggi). Fluctuating salinity was obtained using the

following steps (Figure 2): First, clams were immersed in the initial salinity chambers for 12 h. Second, the net pot was moved to the chamber with the next scheduled salinity treatment. The net pot was then moved to the initial salinity chamber. This procedure was repeated daily. Clams were fed 10 mL of Shellfish Diet 1800® (20,000 cells/mL; Reed Mariculture Inc., Campbell, CA, USA) twice a day.

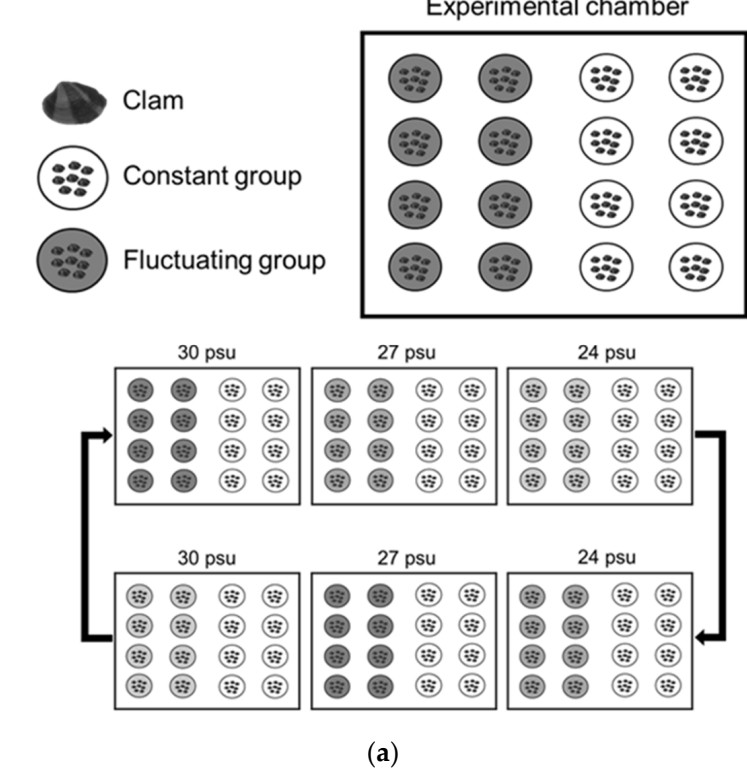

(a)

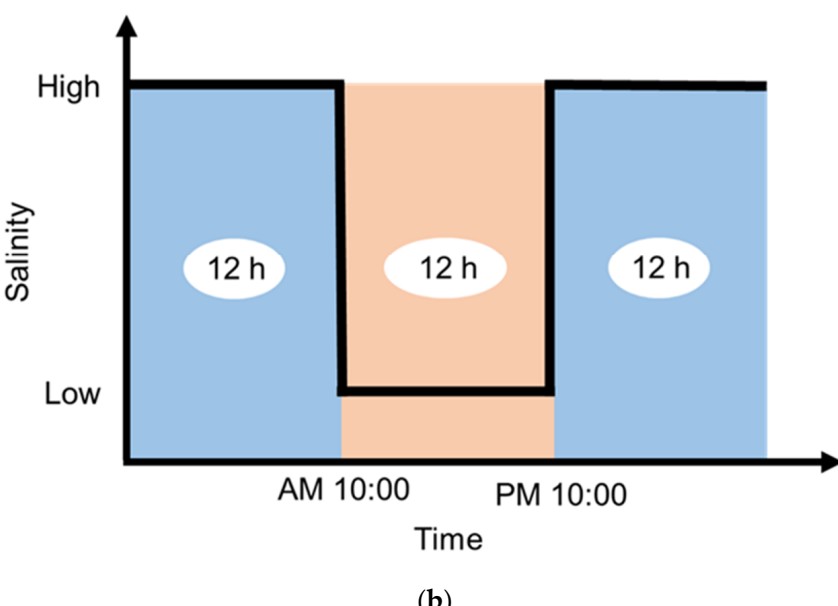

(b)

**Figure 2.** (**a**) Schematic diagram of Experiment 2 setup and (**b**) an example of the salinity fluctuation procedure of Experiment 2.

### 2.2.3. Temperature and Salinity (Experiment 3)

The results of Experiment 1 and Experiment 2 showed no significant difference in mortality between 21 °C and 24 °C, or between 30, 27, and 24 psu (see the Results). Thus, to investigate the combined effect of temperature and salinity on *V. philippinarum*, we conducted an additional experiment based on these results from 10 October to 17 October 2020. A total of 576 clams (SL 34.33 ± 0.15 mm; mean ± SE) were randomly selected from the acclimation tank where they were maintained under steady conditions for eight days (*T* = 21 °C; Sal = 24 psu). Eight clams were allocated to each replicate. Twelve replicates per treatment were placed inside each experimental chamber. The clams were maintained under six experimental conditions for eight days. We exposed clams to the following experimental treatments with two different levels of temperature and three different levels of salinity: (1) Low temperature and low salinity (LTLS, 21 °C, and 18 psu); (2) low temperature and middle salinity (LTMS, 21 °C, and 21 psu); (3) low temperature and high salinity (LTHS, 21 °C, and 24 psu); (4) high temperature and low salinity (HTLS, 24 °C, and 18 psu); (5) high temperature and middle salinity (HTMS, 24 °C, and 21 psu); and (6) high temperature and high salinity (HTHS, 24 °C, and 24 psu). The temperature and salinity treatment values were selected according to the range of natural variations of those parameters observed at the holding facility thereabouts at the rearing site (Sea Grant Gyeonggi). The clams were fed 10 mL of Shellfish Diet 1800® (20,000 cells/mL; Reed Mariculture Inc.) twice a day.

### 2.3. Measurements

Mortality, condition index, and valve closure were measured in Experiments 1 and 2. Mortality and valve closure were measured in Experiment 3.

### 2.3.1. Mortality

The number of dead individuals was checked in each jar at 10:00 and 22:00 daily during the experimental period. A clam was determined as "dead" when it maintained the gaping valve or showed decreased closure power of the shell despite mechanical stimulus [9,41]. Mortality was calculated as follows:

$$Mortality\ (\%) = \frac{No.\ of\ dead\ individuals}{Original\ no.\ of\ individuals} \times 100$$

### 2.3.2. Condition Index (CI)

The condition index (CI) is an ecological and physiological indicator that is widely used to assess bivalve fitness and the effects of environmental stresses [21,36]. After each experiment, separated weights for tissues and shells were obtained after dissecting and drying in an oven (VS-1202D3-S, Vision, Bucheon, Korea) at 65 °C for 48 h and then weighted using an electronic microbalance (PX224KR/E, Ohaus, NJ, USA). CI was calculated according to the formula provided by Walne (1976) [42]:

$$CI = \frac{Dry\ tissue\ weight\ (g)}{Dry\ shell\ weight\ (g)} \times 100$$

### 2.3.3. Valve Closure

During the experimental period, the number of clams closing their valves in each jar was checked daily at 09:00 and 21:00. The percentage of individuals with valve closure was calculated as follows:

$$Valve\ closure\ (\%) = \frac{No.\ of\ valve\ closure\ individuals}{No.\ of\ individuals\ in\ each\ replicate} \times 100$$

### 2.4. Statistical Analysis

Arcsine square root transformations were performed on all proportional data (mortality and valve closure rate of clams). In Experiments 1 and 2, we conducted Kruskal–Wallis *H* test to determine whether the mortality and condition index was influenced by temperature or salinity. If there was a significant difference in the Kruskal–Wallis *H* test, Dunn's test was performed as a post-hoc test for multiple comparison analysis. When there was no significant difference in the Kruskal-Wallis *H* test, however, Mann-Whitney *U* tests were used to detect the significant difference between the two treatment groups independently (e.g., 15 °C vs.18 °C, 15 °C vs. 21 °C, 15 °C vs. 24 °C, 18 °C vs. 21 °C, 18 °C vs. 24 °C, and 21 °C vs. 24 °C). In Experiment 3, a two-way ANOVA was used to determine whether mortality was influenced by combining the two temperatures and three salinities. In all experiments, a repeated-measures ANOVA was applied to determine the effects of each factor on valve closure because not only the independent variable (temperature, salinity, or combining temperature and salinity), but also time can affect valve closure. If there was a significant difference, Tukey's post-hoc test was applied. When sphericity (equality of variance) was violated (Mauchly's test, $p < 0.05$), Huynh–Feldt corrections were applied. Statistical analyses were conducted using SPSS ver. 19.0 (IB8M Corp., Armonk, NY, USA).

## 3. Results

### 3.1. Temperature (Experiment 1)

There was no significant difference in mortality between treatments (Kruskal–Wallis *H* test; $x^2 = 6.034$, df = 3, $p = 0.110$; Figure 3a). However, mortality was significantly different between the 18 °C and 24 °C conditions (two-tailed Mann-Whitney *U* test; 18 °C vs. 24 °C: $U = 17.5$, $n_1 = 9$, $n_2 = 9$, $p = 0.032$). The mortality rate at 24 °C was the highest. There was no significant difference in CI between different temperatures (Kruskal–Wallis *H* test; $x^2 = 5.034$, df = 3, $p = 0.169$; Figure 3c). There was a significant difference in valve closure between treatments (repeated measures ANOVA; $F_{3, 32} = 3.446$, $p = 0.028$; Figure 3b). Individuals were significantly more inactive at 18 °C than at 24 °C (Tukey's post hoc test; $p = 0.018$).

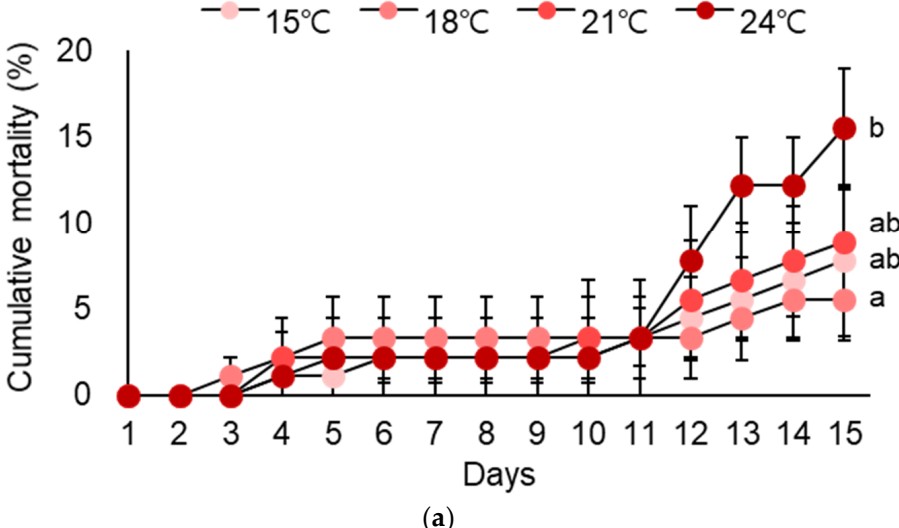

(a)

**Figure 3.** *Cont.*

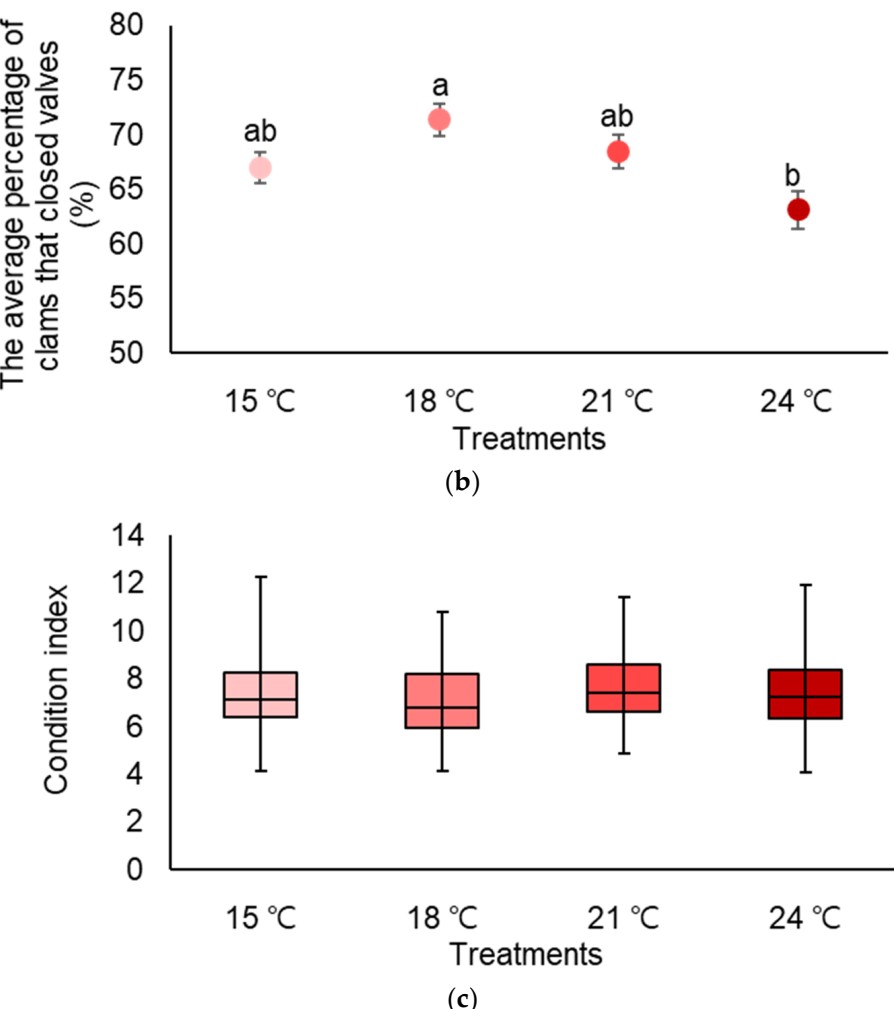

**Figure 3.** (**a**) Cumulative mortality rate (mean ± SE), (**b**) the average proportion of valve closure (mean ± SE), and (**c**) condition index (mean ± SE) of *Venerupis philippinarum* during exposure to different temperatures (15 °C, 18 °C, 21 °C, and 24 °C) for 15 days. Different letters indicate significant differences between treatments ($p < 0.05$).

### 3.2. Salinity (Experiment 2)

There was a significant difference in mortality between treatments (Kruskal–Wallis *H* test; $x^2 = 12.513$, df = 5, $p = 0.028$; Figure 4a). Although the mortality under constant conditions did not significantly differ between the various treatments (24 vs. 27 vs. 30 psu), mortality under fluctuating conditions did differ significantly between 24–30 psu and 30–27 psu (Dunn's test; $p = 0.016$). In addition, the mortality of individuals exposed to two fluctuating salinity conditions (24–30 psu, 27–24 psu) was higher than those exposed to a constant 30 psu (Dunn's test; 24–30 psu vs. 30 psu: $p = 0.005$; 27–24 psu vs. 30 psu: $p = 0.033$). There were no significant differences in CI between the different treatments (Kruskal–Wallis *H* test; $x^2 = 2.272$, df = 5, $p = 0.810$; Figure 4c). There was a significant difference in valve closure between treatments (repeated measures ANOVA; $F_{5,42} = 4.713$, $p = 0.002$; Figure 4b). The valve closure under constant conditions was significantly different between 24 and 30 psu (Tukey's post hoc test; 30 psu vs. 24 psu: $p = 0.018$). Valve closure under fluctuating conditions showed no significant differences between different treatments (24–30 vs. 27–24 vs. 30–27 psu). Clams were significantly more active when exposed to constant salinity of 30 psu than to constant salinity of 24 psu or fluctuating salinity conditions (24–30 and 27–24 psu) (Tukey's post hoc test; 30 psu vs. 24–30 psu: $p = 0.022$; 30 psu vs. 27–24 psu: $p = 0.021$).

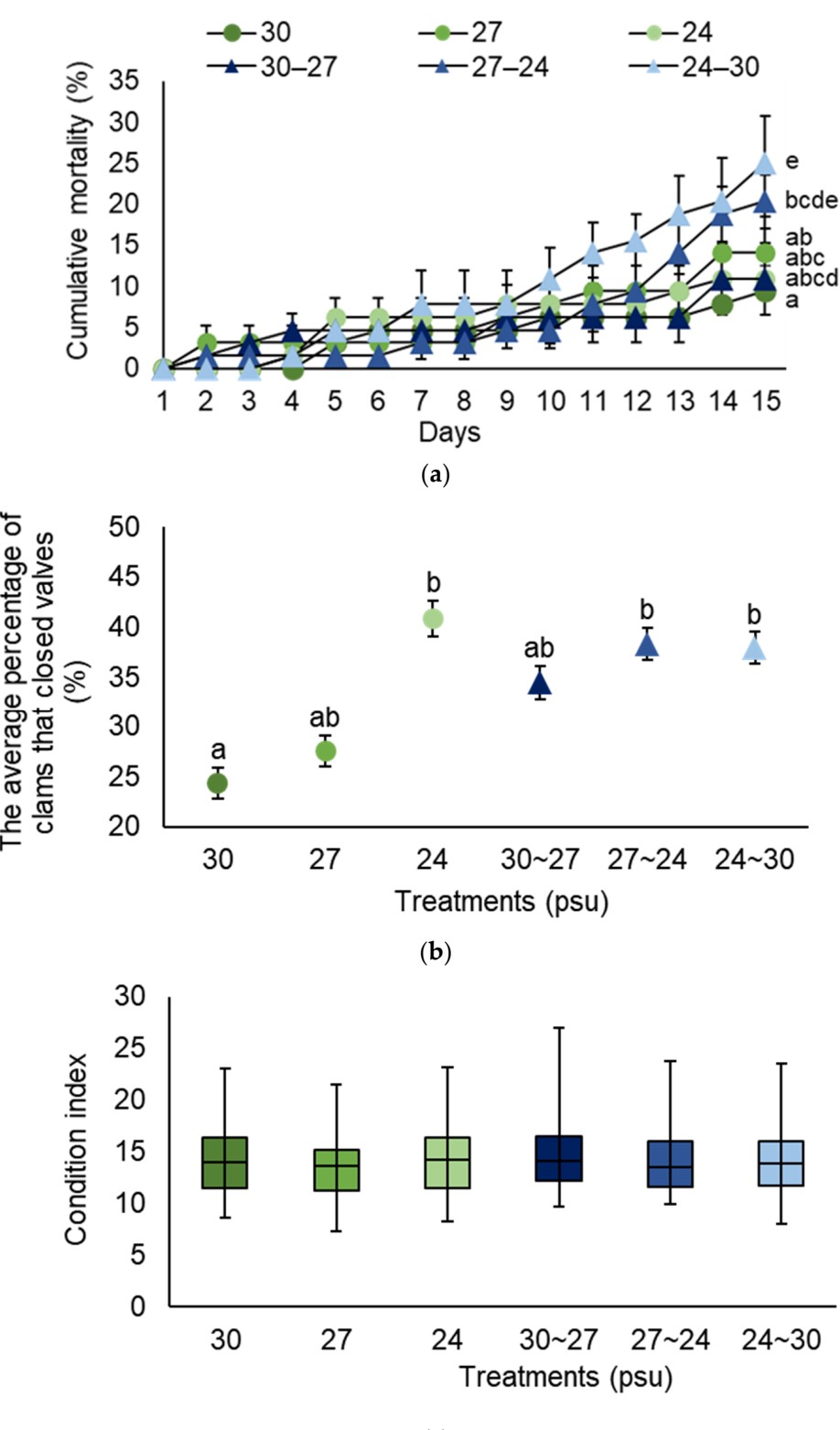

**Figure 4.** (**a**) Cumulative mortality rate (mean ± SE), (**b**) the average proportion of valve closure (mean ± SE), and (**c**) condition index (mean ± SE) of *Venerupis philippinarum* during exposure to different salinities (constant condition: 30, 27, and 24 psu, fluctuating condition: 30–27, 27–24, and 24–30 psu) for 15 days. Different letters indicate significant differences between treatments ($p < 0.05$).

### 3.3. Temperature and Salinity (Experiment 3)

Temperature, salinity, and the interaction between the two factors significantly influenced mortality (two-way ANOVA, temperature: $F_{1, 66}$ = 36.690, $p$ < 0.001; salinity: $F_{2, 66}$ = 3.533, $p$ = 0.035; temperature $\times$ salinity: $F_{2, 66}$ = 4.099, $p$ = 0.021; Figure 5a). After eight days of exposure, mortality significantly increased at a water temperature of 24 °C and at a salinity of 18 psu. The highest mortality was observed under the combination of high temperature (24 °C) and low salinity (18 psu). Temperature, salinity, and the interaction between the two factors also significantly influenced valve closure (Figure 5b and Table 1). The valve closure was significantly lower at a water temperature of 24 °C and at a salinity of 18 psu than that at low temperatures (21 °C) and/or high salinity (24 psu). The lowest valve closure was observed under the combination of high temperature (24 °C) and low salinity (18 psu).

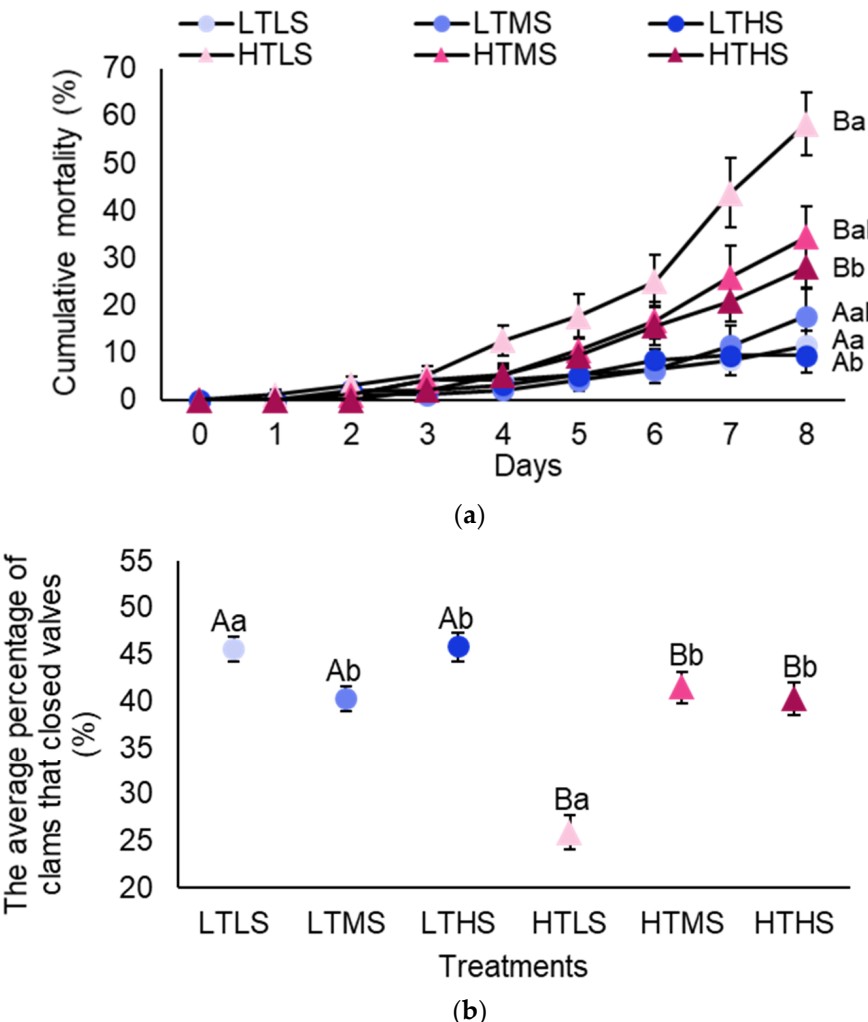

(a)

(b)

**Figure 5.** (a) Cumulative mortality rate (mean $\pm$ SE) and (b) the average proportion of valve closure (mean $\pm$ SE) of *Venerupis philippinarum* during exposure to two different temperatures (21 °C and 24 °C) and three different salinities (18, 21, and 24 psu) for eight days. Different uppercase letters indicate significant differences between treatments for temperature, and different. Abbreviation: LTLS, low temperature and low salinity; LTMS, low temperature and middle salinity; LTHS, low temperature and high salinity; HTLS, high temperature and low salinity; HTMS, high temperature and middle salinity; HTMS, high temperature and high salinity.

**Table 1.** Results of the two-way repeated-measures ANOVA for the effects of temperature and salinity on the valve closure of *Venerupis philippinarum*.

|  | **Valve Closure** |
| --- | :---: |
| Within-subject effect | |
| Day | $F_{380.987,5.773} = 14.900$<br>$p < 0.001$ |
| Day $\times$ Temperature | $F_{380.987,5.773} = 5.674$<br>$p < 0.001$ |
| Day $\times$ Salinity | $F_{380.987,11.545} = 4.542$<br>$p < 0.001$ |
| Day $\times$ Temperature $\times$ Salinity | $F_{380.987,11.545} = 4.932$<br>$p < 0.001$ |
| Between-subject effect | |
| Temperature | $F_{66,1} = 28.418$<br>$p < 0.001$ |
| Salinity | $F_{66,2} = 8.584$<br>$p < 0.001$ |
| Temperature $\times$ Salinity | $F_{66,2} = 16.076$<br>$p < 0.001$ |

## 4. Discussion

### 4.1. Temperature

A reduction in valve closure and increased mortality at 24 °C compared to 18 °C were observed, although the CI of clams showed no significant differences between different temperatures (15–24 °C) for 15 days. The results were similar to those of a previous study on the effects of temperature on juveniles *V. philippinarum* [16]. The increased mortality and behavior change at 24 °C may be a consequence of metabolic depression caused by high temperatures. Han et al. [10] reported that the thermal optimum in scope for growth (SFG) for adults *V. philippinarum* is 20 °C, and Nie et al. [43] found a reduction in oxygen consumption of adults *V. philippinarum* at temperatures >25 °C. The mussel *Mytilus galloprovincialis* has a similar trend when exposed to high temperatures, as shown by the decrease in valve closure at a temperature above 26 °C, the increased mortality, and a reduction in SFG [12,14].

Broadly, bivalves are poikilothermic animals that have a low ability to generate internal temperature through metabolic processes [44], and can maintain survival using low energy compared to that used by homeothermic animals of the same weight. Valve closure is a strategy used under stress [39], which saves metabolic energy and can be implemented under short-term stress [12,20]. However, *V. philippinarum* with aragonite shells, which have higher thermal conductivity, cannot avoid extreme temperatures through valve closure [45,46]. Therefore, temperatures over the optimal range may have a negative effect on *V. philippinarum* in the holding facility.

### 4.2. Salinity

Constant low-salinity conditions did not have a significant effect on mortality or CI; however, constant low salinity had a significant effect on the valve closure of the clams for 15 days. When exposed to salinities near or beyond their tolerance limits, the initial response of bivalves is valve closure. Prolonged valve closing periods may result in mortality, due to increased anaerobic metabolism and reduced filtration [9,20]. However, at salinities higher than 20 psu, the metabolic activities of *V. philippinarum* can recover to the original endogenous rhythm (measured oxygen consumption rate) within 12 h [17]. In addition, several studies have reported that exposure to 24 psu is within the optimal salinity range [23,47]. Nevertheless, Parada et al. [48] reported that long-term exposure to between 25 and 30 psu for approximately 30 days leads to severe mortality in *V. philippinarum* in the field. Our findings indicate that *V. philippinarum* can tolerate seawater at 24 psu

for 15 days; however, more prolonged exposure may lead to the accumulation of effects causing increased mortality.

Previous studies on the effect of salinity fluctuation on *V. philippinarum* showed that salinity above 24 psu did not influence mortality or valve movement [39,49]. For instance, Arisman et al. [49] reported that *V. philippinarum* exposed to 24 psu at 18 °C for 30 days in a 3-h cycle per day did not show different results for mortality and immune response (measure of lysozyme activity) compared to individuals exposed to 34 psu. It was observed that *V. philippinarum* exposed to 25 psu at 17 °C in a 1-day cycle for 15 days did not differ in mortality or valve closure compared to those exposed to 30 psu. However, in this study, clams exposed to 24 psu at 21 °C for 12 h per day showed high mortality and valve closure. This suggests that the salinity fluctuation tolerance range for survival narrowed as the temperature increased, but further information about the metabolic response is needed to understand the combined effects of temperature and salinity fluctuation on the survival and behavior of *V. philippinarum*.

### 4.3. Temperature and Salinity

Temperature and salinity act not only as single stressors, but also as combined stressors. In this study, temperature, salinity, and the interaction between them significantly influenced mortality and valve closure. A reduction in valve closure and increased mortality at 24 °C compared to that at 21 °C was observed. Based on the results of Experiments 1 and 3, the 3 °C temperature increase had no significant effect when clams were exposed to 30 psu. However, when the salinity was less than 24 psu, the range of temperature at which *V. philippinarum* survived was narrowed.

Our findings support those of previous studies in that the temperature range at which the species survived became narrower as salinity decreased [27,50]. In addition, significantly increased mortality and decreased valve closure were observed after exposure to 18 psu compared to that at 21 psu and 24 psu in this study. Unlike in Experiment 2, the reasons for the low valve closure in the high-mortality condition could be that the exposure to 18 psu is close to their tolerance limit. Previous studies have reported that a temperature increase and a salinity decrease may lead to increases in cellular energy expenditure to maintain osmoregulation and induce mortality of hemocyte cells related to the bivalve immune system [51]. Salinity below 20 psu may cause abnormal metabolic activity and decreases the clearance rate [17,52].

Synergistic interactions between multiple stressors are common in nature [29]. In this study, clam mortality was significantly higher under the 24 °C temperature when combined with the 18 psu condition. The interaction between temperature and salinity had a more significant effect on mortality in shorter periods compared to their individual effects in Experiments 1 and 2, respectively, and more than 50% of clams died after eight days of exposure. These results indicate that during the summer period, when extreme climate events occur, the combination of high temperature and low salinity may cause mortalities of *V. philippinarum* in the holding facility within 1–2 weeks.

## 5. Conclusions

Our study provides information on how climate-related events affect the survival and behavior of the infaunal Manila clam *V. philippinarum* in holding facilities. The results showed that a single stressor of temperature or salinity affected behavior and that these effects can be reflected in mortality rates. Furthermore, the combination of warming and low salinity could lead to mass mortality of *V. philippinarum* within eight days. Additionally, the combination of heavy precipitation events and increased temperature may affect the survival and behavior of *V. philippinarum*, and further studies should be conducted on the physiological response of the clams to understand the combined effect of temperature and salinity fluctuations. Based on these results, we recommend that the holding in the summer period should be limited to 1–2 weeks, with the optimal temperature of 18–21 °C and the

optimal salinity range of >24 psu. Moreover, this research can help guide the planning of clam fisheries management and can be used in the design and operation.

**Author Contributions:** Conceptualization, H.B., J.I., S.J., B.C. and T.K.; Methodology, H.B., J.I., S.J. and B.C.; Formal analysis, H.B., J.I., S.J. and B.C.; Investigation, H.B. and J.I.; Writing—original draft preparation, H.B.; Writing—Review & Editing, T.K.; Supervision, T.K.; Funding acquisition, T.K. All authors have read and agreed to the published version of the manuscript.

**Funding:** This research was a part of the projects "Development of 3-D Ocean Current Observation Technology for Efficient Response to Maritime Distress" and "Gyeonggi Sea Grant Research Program" funded by the Ministry of Oceans and Fisheries (grant number 20210642 and 20170362) and was also supported by National Research Foundation (NRF-2020R1A2C1005194) of Korea.

**Institutional Review Board Statement:** Not applicable.

**Informed Consent Statement:** Not applicable.

**Data Availability Statement:** The data that support the findings of this study are available from the corresponding author upon reasonable request.

**Acknowledgments:** We would like to thank the Gunpyeong-ri Fishermen's Association for collecting the Manila clams used in the present study. We thank Jiyeong Choi and Minju Kim for their help with the experimental work. We also thank Seojeong Park, and the lab members provided valuable comments throughout the experiment.

**Conflicts of Interest:** The authors declared no conflict of interests.

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
