# Peer review of "The Effects of Temperature and Salinity Stressors on the Survival, Condition and Valve Closure of the Manila Clam, Venerupis philippinarum in a Holding Facility"

_jmse, doi:10.3390/jmse9070754_

Round 1

Reviewer 1 Report

Review

Paper title: The effects of temperature and salinity stressors on the survival, condition, and valve closure of the Manila clam, Venerupis philippinarum in a holding facility

The Manila clam Venerupis philippinarum is a commercially important species supporting both fisheries and farming in the areas of its distribution. For this reason, it is very important to study the tolerance of this species in different regimes. The authors studied the impact of different combinations of salinity and temperature on some biological parameters of Venerupis philippinarum under controlled conditions. They found that salinity fluctuations were less favorable for this clam as well as higher temperatures. High temperature and low salinity is the most stressful combination for this species. These findings are important to aquaculture and fisheries management of Manila clam and can be used to predict climate-induced changes in marine ecosystems.

All these reasons explain the relevance of the paper by Hyeonmi Bae and co-authors submitted to "Journal of Marine Science and Engineering".

General scores.

The data presented by the authors are original and significant. All conclusions are justified and supported by the results. The study is correctly designed and technically sounds. We authors conducted careful work which will attract the attention of a wide range of specialists focused on the biology of bivalve mollusks and shellfish aquaculture.

Specific comments.

L 19. Change “than of” to “than”

L 43. Change “increased in mortality” to “increased mortality”

L 50. Change “ecological impact” to “environmental impact”

L 69-70. Change “that is an ecologically and commercially important species” to “with high ecological and commercial importance”

L 87. Change “Once in the laboratory, clams” to “In the laboratory, the clams”

L 113-114. Change “temperatures Manila  clams  in  a  holding  tank  can  experience  between  May  and  August  (Sea  Grant  Gyeonggi, Korea).” to “natural seawater temperatures between  May  and  August at the site (Sea  Grant  Gyeonggi, Korea) where the Manila  clams were reared”

L 226. Change “than of” to “than”

L 233. Change “significant difference” to “significant differences”

L 253, 257. Change “a water temperature” to “water temperature”

L 254, 257. Change “a salinity of” to “salinity of”

L 282. Change “a temperature” to “temperature”

L 301. Change “(2012)” to “[47]”

L 304. Change “a more prolonged” to “more prolonged”

L 308. Change “(2017)” to “[48]”

L 310. Delete “those of”

L 339. Change “the summer” to “the summer period”

Author Response

Review

Paper title: The effects of temperature and salinity stressors on the survival, condition, and valve closure of the Manila clam, Venerupis philippinarum in a holding facility

The Manila clam Venerupis philippinarum is a commercially important species supporting both fisheries and farming in the areas of its distribution. For this reason, it is very important to study the tolerance of this species in different regimes. The authors studied the impact of different combinations of salinity and temperature on some biological parameters of Venerupis philippinarum under controlled conditions. They found that salinity fluctuations were less favorable for this clam as well as higher temperatures. High temperature and low salinity is the most stressful combination for this species. These findings are important to aquaculture and fisheries management of Manila clam and can be used to predict climate-induced changes in marine ecosystems.

All these reasons explain the relevance of the paper by Hyeonmi Bae and co-authors submitted to "Journal of Marine Science and Engineering".

General scores.

The data presented by the authors are original and significant. All conclusions are justified and supported by the results. The study is correctly designed and technically sounds. We authors conducted careful work which will attract the attention of a wide range of specialists focused on the biology of bivalve mollusks and shellfish aquaculture.

Specific comments.

L 19. Change “than of” to “than”

Response: We changed it as suggested(Line 19).

L 43. Change “increased in mortality” to “increased mortality”

Response: We changed it as suggested (Line 43).

L 50. Change “ecological impact” to “environmental impact”

Response: We changed it as suggested (Lines 50-51).

L 69-70. Change “that is an ecologically and commercially important species” to “with high ecological and commercial importance”

Response: We changed it as suggested (Lines 70-72).

L 87. Change “Once in the laboratory, clams” to “In the laboratory, the clams”

Response: We changed it as suggested (Line 89).

L 113-114. Change “temperatures Manila  clams  in  a  holding  tank  can  experience  between  May  and  Augus (Sea  Grant  Gyeonggi, Korea).” to “natural seawater temperatures between  May  and  August at the site (Sea  Grant  Gyeonggi, Korea) where the Manila  clams were reared”

Response: We changed it as suggested (Lines 116-119).

L 226. Change “than of” to “than”

Response: We changed it as suggested (Line 224)

L 233. Change “significant difference” to “significant differences”

Response: We changed it as suggested (Line 231).

L 253, 257. Change “a water temperature” to “water temperature”

Response: We changed it as suggested (Lines 251, 255).

L 254, 257. Change “a salinity of” to “salinity of”

Response: We changed it as suggested (Lines 252, 255).

L 282. Change “a temperature” to “temperature”

Response: We changed it as suggested (Line 283).

L 301. Change “(2012)” to “[47]”

Response: We changed it as suggested (Line 302).

L 304. Change “a more prolonged” to “more prolonged”

Response: We changed it as suggested (Line 305).

L 308. Change “(2017)” to “[48]”

Response: We changed it as suggested (Line 309).

L 310. Delete “those of”

Response: We deleted “those of” (Line 311).

L 339. Change “the summer” to “the summer period”

Response: We changed it as suggested (Line 342).

Reviewer 2 Report

The manuscript investigated the effects of temperature and salinity on the survival and activity of Manila clam. There are some questions the reviewer intends to verify:

  1. If the settings of temperature and salinity in this study were similar to the natural environment of Malina clam?
  2. Why the incubation period is 14 days for experiments 1&2 and 7 days for Exp. 3? Exposure time may be also a variable for the survival and activity of the clam.
  3. In Method, why the authors checked the mortality and valve closure at different time points daily? Would this possibly induce the stress of the clam?
  4. Please correct the typo or word spacing (Ex. Line 115, 151, 167, 178, 184, 336…).
  5. In Fig. 5, please describe the abbreviations of LTLS, LTMS, … in the legend.

Author Response

The manuscript investigated the effects of temperature and salinity on the survival and activity of Manila clam. There are some questions the reviewer intends to verify:

  1. If the settings of temperature and salinity in this study were similar to the natural environment of Malina clam?

Response: The experimental water temperature and salinity conditions were set based on the water quality measured between May to August in Gungpyeong-ri, Yellow Sea (Lines 118-119, 130-132, 159-160). However, in the case of salinity fluctuations, it was difficult to reflect the natural environment condition due to the limitations of measuring equipment. Therefore, salinity fluctuation conditions were set based on the salinity data from the Sihwa Lake Tidal Power Station in Yellow Sea and previous studies.

  1. Why the incubation period is 14 days for experiments 1&2 and 7 days for Exp. 3? Exposure time may be also a variable for the survival and activity of the clam.

Response: Kim et al. (2017) showed that juvenile V.philippinarum mortality was higher under a combination of warming and hypoxic stressors than a single stressor after 8 days. Also, in the previous studies, we found that multiple stressors had more negative effects on survival and behavior than single stressors. Based on these, we predicted that the combination of temperature and salinity would have more negative effects than a single stressor. Therefore, the experimental period of experiment 3 was set shorter than experiments 1 and 2. 

  1. In Method, why the authors checked the mortality and valve closure at different time points daily? Would this possibly induce the stress of the clam?

Response: It is difficult to measure at the same time because only one person measures. Also, direct touch only occurred when checking for mortality, and observing valve closure was check only visually. Therefore, there may be no additional stress caused by measuring at different time points daily.

  1. Please correct the typo or word spacing (Ex. Line 115, 151, 167, 178, 184, 336…).

Response: We have corrected the typo or word spacing in the given example lines.

  1. In Fig. 5, please describe the abbreviations of LTLS, LTMS, … in the legend.

Response: We added the information of abbreviations in figure. 5 (Lines 266-269).
